# Attentional shift within and between faces: Evidence from children with and without a diagnosis of autism spectrum disorder

**Eloisa Valenza, Giulia Calignano** *

Department of Developmental and Social Psychology, University of Padova, Padova, Italy

* giulia.calignano@unipd.it

## Abstract

Evidence of attentional atypicalities for faces in Autism Spectrum Disorders (ASD) are far from being confirmed. Using eye-tracking technology we compared space-based and object-based attention in children with, and without, a diagnosis of ASD. By capitalizing on Egly's paradigm, we presented two objects (2 faces and their phase-scrambled equivalent) and cued a location in one of the two objects. Then, a target appeared at the same location as the cue (Valid condition), or at a different location within the same object (Same Object condition), or at a different location in another object (Different Object condition). The attentional benefit/cost in terms of time for target detection in each of the three conditions was computed. The findings revealed that target detection was always faster in the valid condition than in the invalid condition, regardless of the type of stimulus and the group of children. Thus, no difference emerged between the two groups in terms of space-based attention. Conversely the two groups differed in object-based attention. Children without a diagnosis of ASD showed attentional shift cost with phase-scrambled stimuli, but not with faces. Instead, children with a diagnosis of ASD deployed similar attentional strategies to focus on faces and their phase-scrambled version.

## Introduction

Faces recruit infant attention from birth [1,2] but it is during development that the so-called "social brain" emerges through a process of increasing functional specialization [3,4]. Overall, this developmental trend is consistent with an experience-expectant perspective which suggests that both general biases and specific experiences drive functional specialization of face processing, during development [5,6].

Despite similar exposure to faces during early stages of development, it has been consistently reported that individuals with a diagnosis of Autism Spectrum Disorder (ASD) show face-processing atypicalities, including difficulties in deriving and processing socially relevant information from faces [7,8], difficulties in face recognition [9–12], face-discrimination [13], facial expression recognition [14] and eye gaze processing [15]. Such face-processing difficulties might be explained to be the consequence of less time being spent paying attention to

**Data Availability Statement:** All raw data files are available from the Open Science Framework repository (DOI: 10.17605/OSF.IO/JQZ3P).

**Funding:** E.V. grant from Progetti di Eccellenza CARIPARO (rep. no. 1873/2012).

**Competing interests:** The authors have declared that no competing interests exist.

social stimuli by individuals with a diagnosis of ASD, compared with typically developing (TD) controls. This is supported by a metanalysis with 38 eye-tracking studies that analyzed looking fixation times at several social stimuli (i.e. the eyes, the mouth, the face, the body) and non-social stimuli (i.e., the non-social elements and the whole screen) [16]. The results suggested the presence of atypical attention allocation in individuals with ASD, indicated by a reduced attention to the eyes, the mouth and the face and an increased attention to the body and the non-social elements. However, not all of the studies confirm these patterns of data. For example, some studies reported that children and young adults with a diagnosis of ASD prioritize social stimuli to the same degree as TD participants [17–19].

On the one hand, to explain why people with ASD spend less time with attention on faces, some authors have proposed that this might derive from an innate atypicality of the face detection mechanism, that is the subcortical mechanism that tunes infant attention to face-like stimuli from birth [20]. However, evidence indicates that individuals with ASD exhibit entirely typical orienting responses to face-like stimuli, challenging the notion that a primitive sensitivity to protoface stimuli is sufficient for typical social development [21]. Indeed, sensitivity to face-like stimuli is not a sufficient condition for typical social development, and other mechanisms are necessary for the development of the social brain [22]. On the other hand, it has been proposed that atypicalities in orienting visual attention toward faces might not be a specific attentional deficit in ASD. That is, the differences observed between ASD with TD controls might be the result of early atypicalities in the whole attentional network that jeopardizes the emerging "interest" (also and not only) in faces during development [23]. Accordingly, several studies report that individuals with a diagnosis of ASD show attentional atypicalities involving the alerting network: the network responsible for achieving and maintaining a state of sensitivity to incoming information [24,25]. Similar atypicality has been observed in the orienting network: the network responsible for the selection of information from sensory input [26–30]. For example, investigations of children, adolescents, and adults with a diagnosis of ASD have revealed slower, less efficient visual orienting abilities than typically developing (TD) individuals [31]. This evidence points to the presence of core differences in orienting visual attention in individuals with a diagnosis of ASD, that have cascade effects on the attentional mechanism operating in the social domain. However, this conclusion is not supported by a study in which visual attention deployment during passive viewing of images of faces (i.e., human faces, inverted human faces, monkey faces) and objects (i.e., three-dimensional curvilinear objects, two-dimensional geometric patterns objects) was recorded with an eye-tracker system in adolescents with ASD and typical peers. The findings showed that a diagnosis of ASD predicted lower accuracy in face recognition and social-emotional functioning, however, the visual attention patterns between the two groups of participants did not show a substantial difference [32]. In addition, attentional disengagement and social orienting abilities in children with a diagnosis of high-functioning ASD showed no differences compared to those in age-matched and IQ-matched typical developing children [33], as well as in toddlers [34]. Notably, individuals with ASD demonstrate difficulties in the orienting of visual attention toward social images only when they are paired with high autism interest images [35]. Furthermore, social attention in people with ASD seems most impacted when stimuli have high social content (showed more than one person). Therefore, the comparatively low attention deployed towards social stimuli by individuals with a diagnosis of ASD in the context of high social content might be due to difficulty with monitoring higher numbers of events [16]. Altogether these results suggest that differences in attention toward faces in people with, and without, a diagnosis of ASD emerge only under certain conditions and are bound with the task demands.

Given the heterogeneity of the results and interpretations offered by the above cited literature, it becomes relevant to better understand: 1) which conditions trigger different attentional

strategies in children with a diagnosis of ASD compared with children without a diagnosis of ASD, and 2) whether such differences only emerge when individuals pay attention to the face. To answer these questions, we employed a paradigm suitable for evaluating costs and benefits in deploying attention toward different attentional focuses. When the attentional focus of visual orienting is space (i.e., space-based attention), attentional deployment can be thought of as a spotlight moving about the visual field and focusing processing resources on whatever falls within a spatial region, be it an object, a group of objects, or nothing at all [36,37]. In essence, the spatial view of attention suggests that focal attention shifts from one location to another, selecting particular regions in the visual space. Stimuli within these selected regions, regardless of the type of stimulus, are processed more efficiently than stimuli in non-selected regions. Conversely, when the attentional focus of visual orienting is an object (i.e. object-based attention), the target of attention is not an arbitrary region of an unprocessed array, but it is exactly the region that corresponds to candidate objects [38,39]. For object-based attention to be deployed, a robust object representation must be established. Thus, variables that affect the quality of object representations also influence the degree to which object-based attention is employed [40]. That is, space-based and object-based attention lie on a continuum rather than being different discrete components of the orienting of attention. However, unlike space-based attention, only object-based attention is a gateway to investigate the impact of the selected information (object representation) on the deployment of attentional strategies [41].

## The present study

In the current study, we compared the ability to shift attention toward both spatial locations (space-based attention) and objects (object-based attention) in children with and without a diagnosis of ASD.

Space-based and object-based attention has been investigated with a wide variety of paradigms [38,42,43], among these the most popular was developed by Egly, Driver and Rafal [44]. It consists of presenting two stimuli and triggering, with the use of a cue, a participant's attention toward a restricted location of only one of the two objects. Then, a target appears at the cued location (valid condition), or at another location of the cued stimulus (invalid same object condition, ISO condition), or the un-cued stimulus (invalid different object condition, IDO condition). Target detection has been found to be faster for the valid condition compared with the invalid ones, i.e. space-based effect, and in turn, it is faster for the ISO compared with the IDO condition, i.e. object-based effect. This should be an index of object-based attention: when part of an object has received attention the rest of the object benefits perceptually. That is, target detection is facilitated even in the un-cued regions of the object. Note that the distance between the cue and the target, appearing in the cued object, is equal to the distance between the cue and the target that appear in the un-cued object. That is, comparing the ISO with IDO condition, it becomes crucial to control for the distance between cue and target, this rules out the possibility that any difference in target detection is better explained by spatial distance.

Using Egly et al.'s [44] cueing task, Valenza, Franchin and Bulf [45] investigated, in adults and infants, the effect of face and not-face stimuli on both space-based and object-based components of visual attention. The data revealed a cost in target detection (slower saccade latency) for invalid compared with valid conditions both for face and not-face stimuli. These results indicated that attentional selection privileges location-based attention, supporting the well-known benefit of valid cueing for target detection (space-based effect). These findings also put forward the idea that the space-based effect of attention does not differ according to the type of the stimulus, given that target detection was always faster in the valid condition

compared with the invalid conditions, both for face and not-face stimuli. By contrast, the data indicated that object-based facilitation emerges only for not-face stimuli. These findings imply that both adults and not-at-risk infants pay a similar attentional cost when they shift attention within faces or between faces. The authors interpreted this pattern of data as evidence that infants learn very early that faces are relevant and informative stimuli. Consequently when more than one face is present in the visual field, the focus of attention is enlarged to process more efficiently both of the stimuli.

Moving a step forward, the goal of the present study was to investigate how face stimuli and not-face stimuli impact space-based attention and object-based attention in children with, and without, a diagnosis of ASD. Since the selection of a region of space is not affected by the type of object that occupies that region, we should expect no difference between children with, and without, a diagnosis of ASD for the space-based component of visual attention. Conversely, since the selection of an object is affected by the quality of the object representation, then we should expect to observe a difference for the object-based component of attention between the two groups, in particular when the face is the focus of the attentional deployment. In other words, since children with a diagnosis of ASD show difficulties in face processing, then we should expect to observe a substantial difference in the attentional strategies used by children with, and without, a diagnosis of ASD in the face-based component of attention.

## Materials and methods

### Participants

Two groups of children participated in this study. Twenty-five children with a diagnosis of ASD were recruited and tested in two treatment centres of two cities in northern Italy. Inclusion in the ASD group required a previous diagnosis of ASD made by a licensed clinician experienced in the assessment and diagnosis of autism using ADOS-2 (Autism Diagnostic Observations Schedule second edition) or CARS (Childhood Autism Rating Scale). One participant with ASD was excluded from the sample because of low-quality eye-tracking data due to poor calibration of the point of gaze. Ten participants with ASD were excluded because they completed less than 2 valid trials for each level of the design (conditions and stimuli). Thus, the final sample of ASD group comprised fifteen children (10 males, 5 females) with a mean age of 7.6 years (91.58 months, SD = 48 months, range = 46–192 months).

Twenty-four typical development participants were tested in a primary school in a city of north Italy, but only fourteen (9 males, 5 females) with a mean age of 8.7 years (104 months, SD = 42.20 months, range = 48–192 months) were included in the final sample because the others did not match the ASD group in term of age andsex. Typical subjects had no first-degree relatives with an ASD diagnosis. Participants diagnosed with ASD and TD that meet the inclusion criteria were matched on chronological age and sex.

The parents of all participants gave written informed consent for their children before the commencement of data collection. The research protocol performed was approved by our Ethics Committee of the University of Padova, code 1149–2012, the study title is "The role of visual attention in communicative disorders: early predictors of atypical development in high- and low-risk infants". The study was conducted in accordance with accordance with the Declaration of Helsinki.

### Stimuli

Face (F) stimuli and their phase-scrambled version (S) were presented on a black background (see Fig 1). Four women's faces were photographed in a frontal pose with a neutral expression. The photographs were modified with Adobe Photoshop® CS4, and grey faces without hair

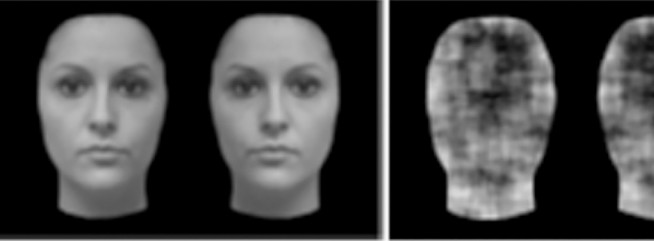

**Fig 1. An example of faces (F), and their phase-scrambled versions (S).**

were generated. For the phase-scrambled versions, these faces were fast Fourier transformed, their power spectra were computed, the phases of the sinusoidal components' waves were randomized, and the inverted fast Fourier transformation was applied [46]. The result was a series of stimuli with different structure and appearance from the original faces, but with the same power spectrum and mean luminance. All of the stimuli measured 10 cm (9.5˚) in width and 15 cm (14.3˚) in height.

The cue and the target were a red dot and a yellow dot, respectively, with a diameter of 3 cm (2.9˚) and a transparency of 41%. The display was virtually divided into five square areas of interest (AOI); one surrounded the central attentional getter (AG) position, and four corresponded to the positions in which the cue and the target could appear. Each AOI measured 4 cm (3.8˚) in width and 4 cm (3.8˚) in height. Equivalent areas of interest were drawn for faces and their scrambled versions.

## Apparatus

The stimuli were presented with E-Prime 2.0 on a 27-inch monitor with a resolution of 1024x768 pixels. A remote infrared eye-tracking camera using bright-pupil technology and placed directly below the monitor was used to collect the data. We used a portable Tobii eye-tracker (Model X2-60 Eye Tracker portable) which recorded the eye movements at a temporal resolution of 60 Hz. This eye-tracking system was mounted on the computer monitor and, therefore, did not interfere with data collection. The system permits head movement, allowing the participants to view in a natural manner.

## Procedure

Testing occurred in a single session in a quiet room at the Centre (for the ASD group) or at the school (for the TD children). The participants sat approximately 60 cm from the monitor. The children were simply told to look at the display and to pay attention. Before beginning the task, point-of-gaze (POG) was calibrated by presenting a looming stimulus in 5 positions of the screen (upper left, upper right, lower left, lower right corners and the centre of the screen) that needed to be reached in order to obtain a reliable calibration. Calibration was made at the beginning of the experimental session. Recalibration only occurred if the participant asked for a pause. Otherwise, the whole experiment relied on the first calibration.

After the calibration procedure, an experimental trial began with a central AG as shown in Fig 2. As soon as the participants looked at the AG, stimuli automatically appeared on the left and on the right side of AG. Participants were presented with two identical adjacent faces or with their phase-scrambled versions as shown in Fig 1. After 1000 milliseconds, a cue superimposed on the top or bottom of one object was presented for 100 milliseconds. The cue presentation was so fast (i.e., 100 milliseconds) that the participants did not have time to plan an

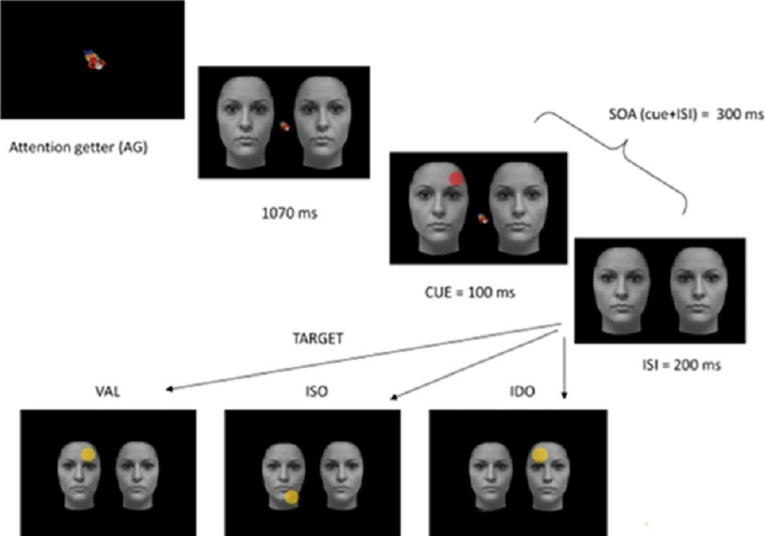

**Fig 2. An example of the three possible target locations (yellow dot): In the Valid (V), Invalid Same-Object (ISO) and Invalid Different-Object (IDO) with respect to the cue's position (red dot).**

overt movement. Immediately after the cue presentation, the AG was removed, and a flashing target appeared automatically after 200 milliseconds. This methodological choice was made to constrain participant attention towards the central point and to prevent an overt movement towards the cue, in the first place. This allowed a fair comparison because, at the cue onset, attention was drawn to the central point in all trials. Importantly, it likely helped to keep the attentional distance constant, not only the actual distance between the cue and target across trials. The target could appear at the cued location (valid target- V), or at the opposite extremity of the cued stimulus, 9cm from the cue (8.6°) (same invalid object- ISO), or in the adjacent un-cued object, 9 cm (8.6°) from the cue (different invalid object- IDO) (Fig 2). The cue and target location probabilities were balanced in the three conditions (33% for each one). The target remained visible until the participants made a saccade toward it, or for a maximum of 2 seconds. The use of the eye-tracker allowed us to control the eye position coordinates during all phases of the experiment and to eliminate the trials in which the participants moved their eyes from AOI corresponding to the central fixation point during the cue presentation.

## Design

Each participant saw 48 trials with four possible cue/target positions (up-right, up-left, down-right, down-left) in 3 conditions (VAL, ISO, IDO) x 2 stimuli (F, S). We also presented four pauses during the experimental session (every 12 trials) with a cartoon video that captured the participants' attention. After completing the cartoon video, the experimenter proceeded with the following trials if the participants were paying attention to the monitor. As soon as the participants became inattentive, the experimenter could stop the experimental session at any moment and restart it as soon as the participant paid attention again. We presented in random order stimuli (F, S) and conditions (V, ISO, IDO). The whole experiment lasted about 10–15 minutes.

## Data analysis

To compute the benefit/cost in terms of time for target detection in the 3 conditions (VAL, ISO and IDO), we measured saccade latency that is defined as the time between the onset of

the target and the first saccade which falls within the target AOI (i.e., saccade onset-target onset). We included in the analysis only those trials that met four quality criteria: (a) at target onset the participants' gaze was found at central AOI, (b) saccade latency lasted longer than 100 milliseconds (i.e., early movements were rejected), (c) saccade latency lasted less than 2 seconds after the onset of the target, (d) participants' gaze entered the AOI that contained the target. Importantly, we included age as a continuous predictor in all models.

Because target detection time was distributed with positive skewness and heteroscedasticity, we used Generalized Linear Mixed effect Models (GLMMs) [47]. GLMMs account for random and fixed effects and have been implemented in similar developmental studies to account for eye-tracking measures [48]. GLMMs with Gamma family and log link function [49] were used to test if the CONDITION and STIMULUS as independent variables predicted target detection time within each group. We compared 6 different models using the lme4 package to select the best approximation [50] for the data of both groups. We included the null model (i.e., target detection time regressed on by-participant random slope for conditions) and proceeded by adding predictors. Akaike Information Criterion weight, which compares all models at once, was used as an index of goodness of fit. Given different AICs for different models, the one with the lowest delta AIC value and higher weight is preferred.

## Results

Fig 3 shows the mean target detection time for each trial across conditions and stimuli. The density distribution of target detection times for conditions, stimuli and each AOI is shown in S1 Fig.

As shown in Table 1, the M6 model best fitted target detection time data (S2 Fig) i.e. three-way interaction term Condition * Stimulus * Group + age.

We estimated linear regression coefficients, 95% CIs and approximated p-values (Table 2) of the M6 model. The selected interactive model shows shorter target detection time for the VAL condition (computed as reference) than for the invalid conditions, independent from the presentation of face and not-face stimuli in both groups. Moreover, ASD predicted slower target detection time in the valid condition compared with the TD group.

The three-way interaction revealed that, in the TD group emerges an attentional shift cost, in terms of longer saccade latency, triggered by the IDO compared with the ISO condition, but only when phase-scrambled stimuli were presented. Accordingly, face stimuli triggered a benefit in shifting attention between-objects. In the ASD group, no cost for target detection time emerged by comparing the IDO with the ISO condition (Fig 3). That is, phase-scrambled stimuli did not trigger any attentional shift cost between- and within-objects, compared with face stimuli, as was the case with the TD group.

In summary, both groups benefitted from the VAL condition compared with the invalid conditions. However, in contrast to the TD group, in the ASD group no cost emerged in attention shift between- vs within-objects for both face and phase-scrambled stimuli (Table 2). Finally, age (treated as a continuous variable) did not show to predict any substantial effect. This result suggests that the effects found in our study likely detected a difference at the group level because our analysis is suitable to estimate effects at both group and participant levels, in an object-based attention task [51].

## Discussion

We extend the investigation of the role played by face and not-face stimuli in both space-based and object-based attention in children with, and without, a diagnosis of ASD. Atypicalities in orienting visual attention have been widely documented in studies on ASD [23,26–28,30],

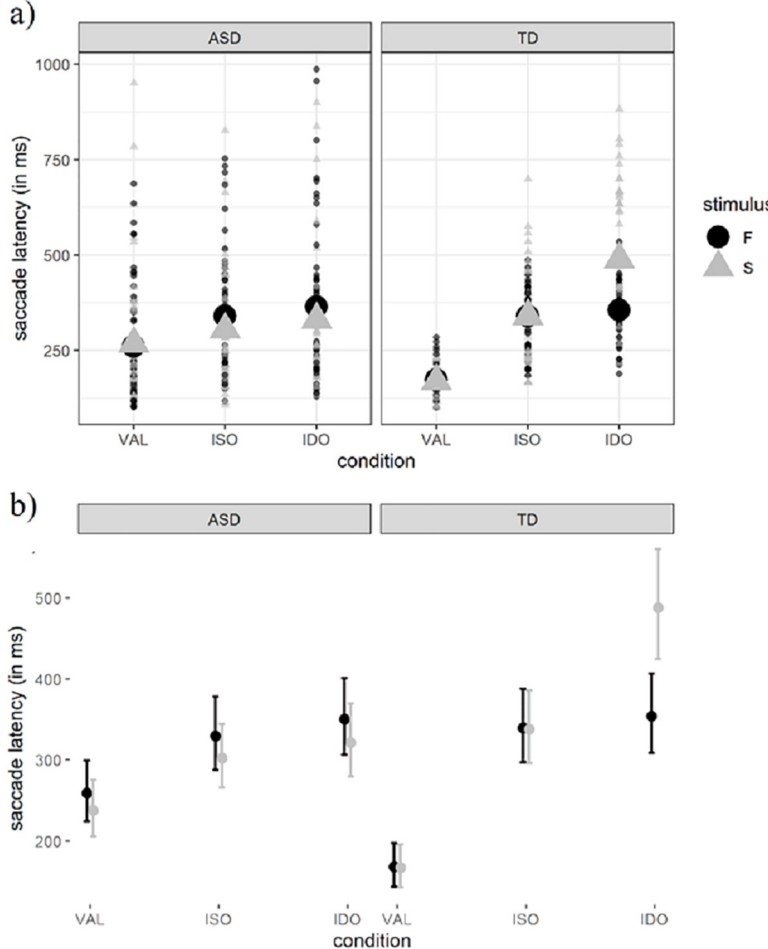

**Fig 3.** a) Average of the target detection time for the Face (F) and the phase-scrambled stimulus (S) indicated by the black circle and gray triangle respectively, across the three conditions (VAL, ISO, IDO) and groups (TD and ASD). Small dots stands for target detection time at the trial level. b) Marginal effects of interaction terms of the selected model (M6) for target detection time in milliseconds; group, i.e. ASD and TD; stimulus i.e. Face (F) and phase scrambled stimuli (S); conditions, i.e. VAL, ISO and IDO.

especially those toward social stimuli [9,31,52,53]. However, to our knowledge, this is the first attempt to test children with a diagnosis of ASD by capitalizing on an attentional task that probes both space-based and object-based components of attention.

**Table 1. The model selection for target detection time in milliseconds.**

| Model | RD | dAIC | AICw | $\eta^2$ |
|---|---|---|---|---|
| **M0.** target detection time ~ (condition\|participant) | 566 | 180.2 | .00 | \ |
| **M1.** target detection time ~ age + (condition\|participant) | 554 | 94.2 | .00 | \ |
| **M2.** target detection time ~ condition + age + (condition\|participant) | 552 | 33.7 | .00 | .010 |
| **M3.** target detection time ~ stimulus + age + (condition\|participant) | 553 | 24.5 | .00 | .000 |
| **M4.** target detection time ~ group + (condition\|participant) | 553 | 22.6 | .00 | .000 |
| **M5.** target detection time ~ condition + stimulus + group +age + (condition\|participant) | 550 | 4.6 | .09 | .005 |
| **M6.** target detection time ~ condition * stimulus * group + age + (condition\|participant) | 543 | 0.0 | .91 | .003 |

dAIC = differential Akaike Information Criterion, AICweight and the Residual deviance. All models included the random effects of participants.

**Table 2. Estimated linear regression coefficients,95% confidence intervals and p-value associated with each predictor of the best-selected model M6.**

| | Target.Detection.Time | | |
|---|---|---|---|
| *Predictors* | *Estimates* | *CI* | *p* |
| (Intercept) | 251.46 | 209.19–302.27 | <**0.001** |
| condition [ISO] | 1.27 | 1.05–1.54 | **0.013** |
| condition [IDO] | 1.35 | 1.13–1.62 | **0.001** |
| stimulus [S] | 0.92 | 0.78–1.08 | 0.305 |
| group [TD] | 0.65 | 0.52–0.81 | <**0.001** |
| age | 1.00 | 1.00–1.00 | 0.624 |
| condition [ISO] * stimulus [S] | 1.00 | 0.80–1.26 | 0.997 |
| condition [IDO] * stimulus [S] | 1.00 | 0.80–1.25 | 0.997 |
| condition [ISO] * group [TD] | 1.58 | 1.20–2.08 | **0.001** |
| condition [IDO] * group [TD] | 1.55 | 1.19–2.03 | **0.001** |
| stimulus [S] * group [TD] | 1.08 | 0.84–1.39 | 0.533 |
| (condition [ISO] * stimulus [S]) * group [TD] | 1.00 | 0.72–1.40 | 0.982 |
| (condition [IDO] * stimulus [S]) * group [TD] | 1.39 | 1.00–1.93 | 0.052 |
| **Random Effects** | | | |
| $\sigma^2$ | 0.15 | | |
| $\tau_{00\ id}$ | 0.03 | | |
| $\tau_{11\ id.conditionISO}$ | 0.03 | | |
| $\tau_{11\ id.conditionIDO}$ | 0.03 | | |
| $\rho_{01}$ | -0.69 | | |
| | -0.60 | | |
| ICC | 0.13 | | |
| $N_{id}$ | 29 | | |
| Observations | 563 | | |
| Marginal $R^2$/Conditional $R^2$ | 0.310/0.402 | | |

The marginal R-squared = the fixed effects variance; conditional R-squared = the variance of the fixed and random effects. The number of observations for the analysis is also reported. The p-value is based on the t-statistics and the normal distribution.

The space-based component of attention measures target detection time in response to different cue conditions that may be valid or invalid. The comparison of target detection time on valid and invalid trials allows the investigation of whether cues direct attention to a particular region, benefiting the processing of the stimulus within the selected region, compared with the processing of a stimulus in a non-selected region. Crucially in all conditions stimulus processing is not influenced by the type of the stimulus, but only by the position of the cue and that of the target. We found a cost to shift attention in the invalid condition compared with the valid condition in both groups. Our findings replicate and extend those obtained in previous studies revealing that even children with a diagnosis of ASD showed a spatial facilitation effect when the cue and the target appeared in the same location. Interestingly, this result, which occurs with few trials and a small sample, suggests that a basic attentional mechanism (the space-based component of attention) might work efficiently in children with a diagnosis of ASD as well as in TD children.

Nevertheless, we observed a substantial difference in the attentional strategies used by children with, and without, a diagnosis of ASD in the object-based component of attention. In particular, in contrast to space-based attention, object-based attention is affected by object representation, meaning that, variables that affect the quality of object representations also influence the degree to which object-based attention is utilized [40]. Accordingly, we

registered a different pattern of data for children with, and without, a diagnosis of ASD. More specifically, for TD children we obtained an object-based effect (i.e. a cost to shift attention within- vs between-objects emerged), but only when the target appeared on the phase-scrambled stimuli. Conversely, when the target appeared on faces, TD children showed the same attentional cost to shift attention within the face or between faces. This evidence replicated previous results, suggesting that in typical development, the object-based component of attention is driven by information prompted by specific stimuli [45,54]. By contrast, no object-based cost emerged in the ASD group, suggesting that children with a diagnosis of ASD use similar attentional strategies to shift attention between faces and not-face stimuli. That is, in the ASD group, the space-based components of visual attention mirror those observed in TD children, whereas the object-based components of attention do not. Although our findings support previous results on attentional disengagement and orienting in individuals with ASD [33,34,55], they are inconsistent with the majority of the studies that have compared the time spent paying attention to social stimuli by individuals with or without ASD. This discrepancy might reflect combinations of the following factors.

First, a possible interpretation of our results can be tracked to methodological differences across studies. In our design, we present a task in which attention is driven automatically by a rapid (i.e. 100 ms) and exogenous cue. Similar to previous studies that employed a cueing task [55–57] we found that exogenous orienting in ASD is not as impaired as thought. However, the evidence of a lack of exogenous orienting when more natural cues are adopted (i.e., head and eye gaze, point, clapping hands, calling child's name) suggests that the nature of the cue plays a pivotal role in modulating attentive performances in people with a diagnosis of ASD. A likely interpretation could be that people with autism respond to certain physical features of the cue, such that when these features are removed, atypicality in orienting of attention is no longer observed.

In addition to the type of cue, also the information prompted by the specific stimuli dramatically influence orienting of attention. Most of the previous studies have used competing stimuli to evaluate which of the two stimuli trigger more attention in children. However, using a preferential viewing task, Unruh et al. [35] have recently demonstrated that atypicality in social attention in individuals with a diagnosis of ASD may be context-dependent. Indeed, in that study adolescents with ASD exhibit longer latency compared with TD participants while orienting attention to faces paired with high autism interest images (i.e., trains, vehicles, aeroplanes, clocks). Notably, in that study adolescents with a diagnosis of ASD did not replicate this cost when social stimuli were paired with low autism interest images (i.e., clothing, tools, musical instruments, plants). This evidence stresses the importance of carefully choosing non-social stimuli to clearly disambiguate between the impact of social and non-social stimuli on attention, in individuals with ASD. In the present study, we used a paradigm that does not contain competing visual information since only two faces or two non-face stimuli are shown. This methodological choice did not prevent us from comparing the strategies of attentional deployment adopted by participants when presented with faces or their phase-scrambled version. More importantly, the phase-scrambled stimuli were precisely matched with faces in terms of power spectrum and mean luminance, compared to those included in other studies. Thus, we presented stimuli that might reduce the perceptual gap between face versus not-face processing for the ASD group. As a future step, we think that it would be of great interest to investigate-in the context of this paradigm-whether individuals with a diagnosis of ASD show attentional shift cost between-object when presented with high-interest items.

Another methodological difference with previous studies concerns the operationalization of the attentional orienting measure. Data from studies in which the orienting of visual attention has been evaluated by direct observation in a semi-structured face-to-face interaction

[29,53] contrast with those in which orienting of attention has been explored through experimental paradigms and gaze-tracking technology. As with most of the recent studies [33–35], we used precise oculometric measures. The use of an infrared eye-tracking system yields more accurate data than those produced in studies in which the attentional abilities are calculated from the videotaped recordings of the children's eye movements, and it should be preferred in future research.

Furthermore, it is fundamental to outline here that we observed greater variability in data from the ASD group compared to the TD group (Fig 3). Importantly, data quality and total number of trials per group did not explain such difference hence, the observed heterogeneity is likely to speak for the high variability usually found in individuals with a diagnosis of ASD compared with TD controls [58,59]. Importantly, in contrast to classical statistical analysis selected in most of the studies cited above, e.g. ANOVAs, our statistical analysis accounted for individual variability, i.e. random slope, while estimating the effects of interest. That is, ANOVAs are not able to estimate if a single participant or a sub-group of participants is driving the effect. The GLMs help to control for individual variability, which are considered as random effects i.e., random slope, in a way that, statistical estimates account for individual differences and allow us to better analyze the likeliest effect at the group level. Even more critically, ANOVAs assume that each observation, i.e., trial, is independent, which is not the case with a repeated measures design. Thus, we encourage future studies, in particular, those interested in a population characterized by a wide heterogeneity, as is the case of ASD, to select those statistical approaches that account for individual variability. This methodological choice will help to move toward better profiling of so-called "atypical" cognitive outcomes.

Second, one could also argue that when faces are involved (wherever the cue position is), it is required to consider specific regions of the face (i.e., the eye or mouth regions) that may or may not preferentially capture the viewer's attention. According to a large amount of data about eye avoidance/aversion effect, this could show a weakness of the study. Nevertheless, data presented in S3 Fig. showing target detection times split for AOIs, i.e. eyes and mouth region (for face and not-face stimuli) and, both a meta-analysis of 38 studies [16] and a literature review on social attention in individuals with ASD [31] weaken the assumption that individuals with ASD demonstrate an excess of attention on the mouth and diminished attention on the eyes compared with TD individuals. The present study suggests that it is worth analyzing the effect prompted by the whole face because it offers a better account of face processing and recognition abilities [60]. Moreover, estimating the impact of the whole face on attentional strategies provides evidence of a selective social deficit in ASD. Indeed, if individuals with ASD have a bias towards avoiding or scarcely processing social stimuli, there is no theoretical need to split the face into sub-components. Separate sub-components reduce the reliability of any definition of social stimulus and the likelihood of capturing differences in face recognition abilities [60–62]. That is, a single sub-component, e.g., the eyes, does not talk about impairments in social orienting.

Nonetheless, our results do not rule out the possibility that deficits in orienting visual attention toward faces may emerge under real-world conditions, even if they are not apparent in more constrained laboratory tests. The findings by Dawson et al. [63], for example, point toward this possibility. In their experiment, conducted during face-to-face interaction with children with ASD, social and non-social auditory stimuli (e.g., humming and snapping fingers vs a phone ringing or blowing a whistle) were produced by one experimenter while another experimenter was interacting with the child. Children with ASD were less likely than TD children to orient their attention toward social sounds; this effect reduced for non-social sounds.

In addition, a factor that may help to explain our results concerns the demographic characteristics of participants such as their mental and chronological age. It is well-known that ASD is a disorder with a vast heterogeneity, and it has been shown that mental age (both verbal and nonverbal) is a significant predictor of ASD cognitive performances, including attentional abilities. In a study, adults with a diagnosis of ASD and a low IQ (~40) performed a gap-overlap task showing slower disengagement of attention in the overlap condition, compared with TD controls [64]. Remarkably, an IQ-matched group showed a similar disengagement delay to the ASD group. Viceversa, Fischer et al. [33] demonstrated that children with a diagnosis of high-functioning ASD do not suffer from impairments in attentional disengagement. Exogenous and endogenous orienting, as well as gaze cueing, appear intact in children with a diagnosis of high-functioning ASD [55]. In the present study, ASD and TD groups were matched only on chronological age and sex. The only information associated with the level of mental age in the ASD group regards general low verbal (vs nonverbal) communication skills. The lack of a match of IQ or mental age between the groups that participated in the present study requires future investigations. In particular, future studies should weight the role played by mental age in predicting differences at the group and individual level in space- and the object-based attention.

In summary, the heterogeneity of research designs (i.e., the attentional mechanisms explored, the type of cue and stimuli adopted, the measure of visual orienting, the setting of observations) may explain the far from being confirmed set of findings reported in the vast literature on visual orienting to face in people with ASD. We encourage investigations like the present study, also in clinical assessment, because they offer objective tools and useful measures that help to disentangle which attentional strategies are activated (at the individual and the group level) for face and not-face-stimuli, by individuals with a diagnosis of ASD.

## Conclusions

Having reviewed the literature on eye-tracking and the orienting of attention in individuals with, and without, a diagnosis of ASD, we proposed an attentional probe task as both a research and clinical opportunity to analyze clear and consistent eye movement patterns, suitable for tracking strategies of attention deployment across individuals and groups. We believe that the strength of this task is that it is suitable for measuring both strategies of attention deployment that work whatever the type of object on which attention is focused (space-based attention), as well as strategies of attention deployment affected by the nature of the object (object-based attention). More simply the task used in this study allows us to investigate immediately two basic components of attention that may, or may not, be affected by face processing. The findings of this study indicate that in contrast to typically developing controls, children with a diagnosis of ASD deploy similar attentional strategies to focus on faces and their phase-scrambled version.

## Supporting information

**S1 Fig. Frequency density plot.** Target detection times in milliseconds per group i.e. TD and ASD, stimulus i.e. phase scrambled stimuli (S) or Face (F) and conditions, i.e. VAL, ISO and IDO.
(TIF)

**S2 Fig. Fit of residuals.** Residual distribution for detection times (in milliseconds) to non-censored data i.e. Gamma, by maximum likelihood (mle).
(TIF)

**S3 Fig. Frequency density plot.** Target detection times in milliseconds per group i.e., TD and ASD. Moreover, considering four AOIs i.e., eye = upper face, mouth = lower face, not-face eye = upper scrambled-phase, not-face m = lower scrambled-phase stimulus. Plots reflect only a sub-sample of participants (12 TD and 7 ASD) that reached at least three valid trials for each AOIs (in each condition and for each stimulus).
(TIF)

## Acknowledgments

We would like to thank all the children and families who participated in our study. We also thank Terence de Michele, Francesca Abalti and Margherita Maran for their assistance with data collection.

## Author Contributions

**Conceptualization:** Eloisa Valenza.

**Formal analysis:** Giulia Calignano.

**Funding acquisition:** Eloisa Valenza.

**Investigation:** Giulia Calignano.

**Methodology:** Eloisa Valenza, Giulia Calignano.

**Visualization:** Giulia Calignano.

**Writing – original draft:** Eloisa Valenza.

**Writing – review & editing:** Giulia Calignano.

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
