## [Decision Letter · Decision Letter 0]

8 Dec 2020

PONE-D-20-35397

Attentional shift within and between faces: evidence from children with and without a diagnosis of autism spectrum disorder

PLOS ONE

Dear Dr. Calignano,

Thank you for submitting your manuscript to PLOS ONE. After careful consideration, we feel that it has merit but does not fully meet PLOS ONE’s publication criteria as it currently stands. Therefore, we invite you to submit a revised version of the manuscript that addresses the points raised during the review process.

As you can see, both reviewers provided overall positive feedbacks, but also provided numbers of constructive comments. Many of them are minor or stylistic, but there are several major comments about the rationale of the study, details of methodology, data quality and analysis, as well as the points of discussion. I encourage the authors to fully address all the comments of both reviewers, if you intend to submit your revision. 

Among all, Reviewer 1 made a critical observation on your OSF data which requires your urgent action, to protect personal data of the participants.

We look forward to receiving your revised manuscript.

Kind regards,

Atsushi Senju

Academic Editor

PLOS ONE

Journal Requirements:

2. Our internal editors have looked over your manuscript and determined that it is within the scope of our Cognitive Developmental Psychology Call for Papers. The Collection will encompass a diverse range of research articles in developmental psychology, including early cognitive development, language development, atypical development, cognitive processing across the lifespan, among others, with an emphasis on transparent and reproducible reporting practices.  Additional information can be found on our announcement page: https://collections.plos.org/s/cognitive-psychology

If you would like your manuscript to be considered for this collection, please let us know in your cover letter and we will ensure that your paper is treated as if you were responding to this call. Please note that being considered for the Collection does not require an additional peer review beyond the journal’s standard process and will not delay the publication of your manuscript if it is accepted by PLOS ONE.

If you would prefer to remove your manuscript from collection consideration, please specify this in the cover letter. 

4. Please ensure that you include a title page within your main document. We do appreciate that you have a title page document uploaded as a separate file, however, as per our author guidelines (http://journals.plos.org/plosone/s/submission-guidelines#loc-title-page) we do require this to be part of the manuscript file itself and not uploaded separately.

5. Please include captions for your Supporting Information files at the end of your manuscript, and update any in-text citations to match accordingly. Please see our Supporting Information guidelines for more information: http://journals.plos.org/plosone/s/supporting-information

Reviewers' comments:

Reviewer's Responses to Questions

**Comments to the Author**

1. Is the manuscript technically sound, and do the data support the conclusions?

Reviewer #1: Partly

Reviewer #2: Yes

2. Has the statistical analysis been performed appropriately and rigorously? 

Reviewer #1: I Don't Know

Reviewer #2: Yes

3. Have the authors made all data underlying the findings in their manuscript fully available?

Reviewer #1: Yes

Reviewer #2: Yes

4. Is the manuscript presented in an intelligible fashion and written in standard English?

Reviewer #1: No

Reviewer #2: No

5. Review Comments to the Author

Reviewer #1: The study used remote eye-tracking and employed Egly’s paradigm to examine space-based and object-based attention in children with and without ASD. Egly’s paradigm was adapted to show either two identical faces (side by side) or their phase-scrambled versions. The authors reported group differences in object-based – but not space-based – attention: the TD group was faster to detect a target when it was presented in the cued location (i.e., cue and target appeared in the same phase-scrambled stimulus) compared to the un-cued location (i.e., cue and target did not appear in the same phase-scrambled stimulus). The ASD group did not show this pattern. This finding was only evident in the phase-scrambled but not the face condition.

The topic of the paper is multifaceted, the authors employed an interesting experimental paradigm to examine space- vs object-based attention, and the study findings would be of interest to a wide readership. However, some clarifications are required, with my main comments relating to the study motivations, data quality, and wider theoretical implications of the findings.

Major comments

I viewed the csv data sheet in the OSF repository, and the first names of participants were included in the file alongside their age, sex, and ASD diagnosis. The name column urgently needs to be removed. (The readme.txt file also does not seem to correspond with the csv file.)

The authors present relevant literature in their introduction, though the rationale for the present study was still a bit unclear, so that it was not entirely clear why Egly’s paradigm in particular was suitable. For example, while the introduction mentions studies on visual orienting behaviour toward social stimuli in ASD, the current paradigm does not contain competing visual information since only two faces are shown (or their phase-scrambled equivalent). The paper states as an aim “deepen[ing] the understanding of how attentional mechanisms are difunctional in ASD compared with TD in order to offer new useful tools able to hack specific pattern of attention deployment in children with a diagnosis of ASD”, but this should be more specific. It could be helpful to expand on the literature on attentional difficulties in ASD and/or be more explicit about the study aims by elaborating on why space- vs object-based attention specifically requires investigation to better understand social attention in ASD, which would then motivate Egly’s paradigm as well.

Did data quality differ between groups and were there any calibration requirements or data quality metrics used? This could be particularly relevant also for the ASD group, especially if their data quality was lower and led to more excluded trials. For example, Fig3 illustrates much less consistent data patterns for the ASD than TD group, but is this the result of greater heterogeneity in the ASD group or could this also be down to systematic data quality issues?

Related to the point above, there were some critical observations in the data that were not discussed. For instance, the heterogeneity in the ASD group is quite remarkable compared to the TD group (Fig3), but it seems that this was not highlighted anywhere. The ASD group also seems overall slower compared to the TD group, including in the valid condition, which has also not been mentioned (and could also be related to data quality).

Some critical details on the study methodology would need to be included to ensure data integrity.

- It is stated that groups were matched on chronological age, but the ASD group is on average more than 1 year younger – is this an error, how were they age-matched?

- Why were only 14 TD participants included, what happened to the other 10 who were tested?

- What was the motivation to include such a wide age range and collapsing all participants into one group (rather than, e.g., as a continuous measure – this is particularly relevant given that face perception processes are known to change across the tested ages)?

- Did the ASD group consist of children of varying mental ages, is there any information at all?

- How many valid trials did each group actually complete? It states that the criterion of “less than 2 valid trials for each level of the design” was used to exclude ASD participants, but even two trials would not seem enough for analysis. It would be beneficial to see the number of valid trials per group. Also, 48 trials in total for face/scrambled conditions and valid/ISO/IDO – does this mean 8 trials per level?

- Why was cue position a factor in the analysis?

- Why did the ‘four quality criteria’ not confirm that participants made a saccade to the target by examining, e.g., time taken until entry to the AOI that contained the target (rather than the 40px toward the target)?

The discussion mainly focused on methodological differences between the present study and previous studies to explain any discrepancies between findings. While this was of course useful and insightful, a deeper discussion of the theoretical implications would be helpful to highlight the significance of the present findings, e.g. how do group differences in object-based attention specifically relate to or explain social attention in ASD, what may be possible developmental mechanisms? Greater emphasis on such a theoretical discussion is required (and could be added as a final sentence in the abstract).

Minor comments

- A mean effect size of 0.55 was reported (cf. Chita-Tegmark, 2016) but I could not find this value in the original paper

- It might be beneficial to explain space- vs object-based attention earlier in the manuscript.

- “uninterpretable eye movements” is an unusual expression – possibly “low-quality eye tracking data” would work if this is what the authors meant.

- It would help the reader if the text referred to the figures when they are described, e.g. when mentioning the sequence of displays in the methods, refer to Fig2.

- “(POG) was calibrated by presenting an attractive, looming stimulus in 5 positions of the screen […] to validate the accuracy of the calibration” – calibration cannot validate the accuracy; this could be fixed by deleting the part from ‘to validate’

- Figures and tables should use labels that can be understood without the main text. For Fig3: it would be easier to interpret the data if the y-axis for the TD/ASD groups were identical. For Fig4: Please include information on y-axis units (the ticks should also be equally spaced apart), the meaning of error bars, and the nature of the displayed data points (means?). It would help to reduce the 4 plots into 2 by overlaying data if possible. Since x-axis represents categorical data, the data points should not be connected by lines.

- Why is the central attention getter displayed simultaneously with the cue?

- The section “The present study” was not always easy to understand, mostly because statements were quite general (e.g., “deployment of attention and the structure of selected information”, “focal attentional shifting acts upon one discrete object at a time”, “similar attentional cost to shift attention within the cued face and importantly, between the cue and the un-cued face”, “they enlarged the focus of attention in order to monitor all faces”, etc.). Maybe such statements could be more specific.

- The paper would need to be proof-read due to typos/grammatical errors.

Reviewer #2: The authors present a novel application of Egly’s paradigm, first used by Valenza et al. (2014), to compare differences in space- and object-based attention with face versus non-face stimuli, between ASD and TD groups. In a simple and clear task, the authors find a space-based facilitation effect for both groups across both stimulus conditions, and an object-based facilitation effect with face stimuli. They find a difference in performance between the two groups in the IDO condition for phase-scrambled faces: the ASD group, but not TD, demonstrating no object-based cost in attentional shift. The authors suggest that this pattern of results is evidence of autistic groups employing a similar attentional strategy for both face and non-face stimuli, offering a potential route to better understanding of atypical attention to social stimuli often observed in the autistic population. These are an interesting set of initial results, and the finding is a useful addition to the literature. The authors are advised to consult a copy-editor/proof-reader to correct typos and minor grammatical errors. My comments are attached in the document "Review_PONE-D-20-35397.docx".

6. PLOS authors have the option to publish the peer review history of their article (what does this mean?). If published, this will include your full peer review and any attached files.

Reviewer #1: No

Reviewer #2: No

---

## [Author Response · Author response to Decision Letter 0]

11 Feb 2021

Please see the full response to Editor and Reviewers in the CoverLetter.pdf and Response_to_Reviewers.pdf files attached to the submission.

---

## [Decision Letter · Decision Letter 1]

3 Mar 2021

PONE-D-20-35397R1

Attentional shift within and between faces: evidence from children with and without a diagnosis of autism spectrum disorder

PLOS ONE

Dear Dr. Calignano,

Thank you for submitting your manuscript to PLOS ONE. After careful consideration, we feel that it has merit but does not fully meet PLOS ONE’s publication criteria as it currently stands. Therefore, we invite you to submit a revised version of the manuscript that addresses the points raised during the review process.

As you can see, both reviewers are happy with the revision overall, and remaining comments are predominantly minor and stylistic. However, I would strongly encourage the authors to fully address all the comments provided by both of the reviewers.

We look forward to receiving your revised manuscript.

Kind regards,

Atsushi Senju

Academic Editor

PLOS ONE

Journal Requirements:

Reviewers' comments:

Reviewer's Responses to Questions

**Comments to the Author**

1. If the authors have adequately addressed your comments raised in a previous round of review and you feel that this manuscript is now acceptable for publication, you may indicate that here to bypass the “Comments to the Author” section, enter your conflict of interest statement in the “Confidential to Editor” section, and submit your "Accept" recommendation.

Reviewer #1: (No Response)

Reviewer #2: (No Response)

2. Is the manuscript technically sound, and do the data support the conclusions?

Reviewer #1: Yes

Reviewer #2: Yes

3. Has the statistical analysis been performed appropriately and rigorously? 

Reviewer #1: I Don't Know

Reviewer #2: Yes

4. Have the authors made all data underlying the findings in their manuscript fully available?

Reviewer #1: Yes

Reviewer #2: Yes

5. Is the manuscript presented in an intelligible fashion and written in standard English?

Reviewer #1: Yes

Reviewer #2: Yes

6. Review Comments to the Author

Reviewer #1: The revised manuscript is greatly improved, and the authors have addressed my previous comments in a satisfactory manner. The updated structure of the introduction lays out a coherent rationale of the study aims and Egly’s paradigm specifically. The methods are also much clearer. I have a couple of final, very minor comments/suggestions.

- “comorbid attentional atypicalities”: remove ‘comorbid’ since this does not relate to a clinical condition

- Table 2/Fig 3: typo in “Stimulus”

- “challenging the notion that a primitive sensitivity to protoface stimuli is essential for typical social development”: I think the authors meant ‘sufficient’ rather than ‘essential’ (the terms have different meanings). It also seems that the following sentence aims to express the same idea.

- “whether such differences only emerge when attention is focused on the face”: it may be useful to clarify ‘attention is focused on the face’ since this could be interpreted in different ways depending on a reader’s academic background and their interpretation of 'attentional focus'

- Thank you for explaining the age-matching procedure. It sounds like individuals were matched on both age and sex (as also stated in the Discussion), in which case it should be added in the methods for clarification, i.e., in the sentence “Participants diagnosed with ASD and TD that meet the inclusion criteria were matched on chronological age.”

- Thank you for clarifying the quality criteria. It seems the authors adopted my phrasing (“time taken until entry the AOI that contained the target”), but in the context of the ms I would suggest something along the lines of: ‘d) participants’ gaze entered the AOI that contained the target’ - I assume this is what was meant.

- The updated figures are much easier to interpret and highlight the findings, thank you. Are Fig 3 and 4 displaying the same dataset but as individual points (Fig3) vs summary stats + error bars (Fig4)? If yes, I wonder if the graphs can be combined, although not a necessity. Individual points and summary stats are both useful, but if figures show the same data then it could be confusing as to why separate figures were generated.

- “reduce confusion in the estimation…”: not clear what is meant by this part

- “collide with”: contrast?

- “we found a quite remarkable heterogeneity”: for the ms, it might be better written as “we observed greater variability in the data from the ASD group compared to the TD group”, or similar.

- “huge variability”: replace huge with ‘high’

- “Continuos”: typo, ‘continuous’

Reviewer #2: The manuscript is greatly improved. The introduction now reads clearly, and better sets up the premise of the experiment; particularly, the discussion on space-based vs object-based attention and why Egly’s paradigm is a suitable choice for this study. The methods and accompanying figures are more intelligible. The discussion better highlights the implications of the present set of results.

My comments are as follows:

1. Page 2, ref [20] and [21]: “On the one hand, to explain why people with ASD spend less time with attention on faces, some authors have proposed that this might derive from an innate atypicality of the face detection mechanism, that is the subcortical mechanism that tunes infant attention to face-like stimuli from birth [20]. However, evidence indicates that individuals with ASD exhibit entirely typical orienting responses to face-like stimuli, challenging the notion that a primitive sensitivity to protoface stimuli is essential for typical social development [21].”

Please note that reference [20] (Shah, Gaule, Bird, & Cook 2013) proposes that the robust orienting effect towards proto-face stimuli in the ASD population which they find speaks against developmental accounts which suggest that reduced looking behaviour in ASD is a consequence of atypical face detection mechanisms. Accordingly, please cite appropriate references to support the first sentence and please include reference [20] along with reference [21].

2. Page 2: “Accordingly, several studies reported that individuals with a diagnosis of ASD show comorbid attentional atypicalities involving the alerting network”.

Attentional difficulties are not necessarily “comorbid” with ASD in the way that another disorder would be. Please remove “comorbid”, the sentence will then read: … “individuals with a diagnosis of ASD show attentional atypicalities involving the alerting network”.

3. Page 3: “To answer these questions, we employed a paradigm suitable for evaluating costs and benefits in dislocating attention toward different attentional focuses.”

The authors make the argument for space-based and object-based attention lying on a continuum. In which case, might it be more suitable to replace “dislocating” with “deploying” or another similar word.

4. Page 3: “That is, space-based and object-based attention lie on a continuum rather than be different discrete components of the orienting of attention.”

Instead of “be”, “being” might read better.

5. Page 3, last para: “However, unlike space based attention, only object-based attention is a gateway to investigating the complex interplay [of which processes?] underling the impact of the selected information (object representation) on the deployment of attentional strategies”.

“Complex interplay” implies that two or more processes are involved, which is not made clear currently. Is “underling” perhaps “underlying”?

6. Page 4, present study: “In the current study we compared [which groups?]”.

It would be good to state which sample groups were included at the beginning.

7. Page 9, data analysis: “time taken until entry [to] the AOI that contained the target”.

Please include the missing word “to”.

8. Page 11: “However, in contrast to the TD group, in the ASD group no cost emerged in attention shift between- vs within-objects” [for both face and phase-scrambled stimuli].

Perhaps this could this be added to the sentence to summarise results better?

9. Page 13, discussion: “Crucially, [in which trials?], stimulus processing is not influenced by the type of the stimulus, but only by the correspondence between the position of the cue and those of the target.

Please could the authors specify in which trials this correspondence is expected to influence performance, i.e., is this the case for only valid trials or all trials facilitated by space-based attention? Presumably the latter, but this would need to be made clear please.

7. PLOS authors have the option to publish the peer review history of their article (what does this mean?). If published, this will include your full peer review and any attached files.

Reviewer #1: No

Reviewer #2: No

---

## [Author Response · Author response to Decision Letter 1]

17 Mar 2021

6. Review Comments to the Author

Reviewer #1: The revised manuscript is greatly improved, and the authors have addressed my previous comments in a satisfactory manner. The updated structure of the introduction lays out a coherent rationale of the study aims and Egly’s paradigm specifically. The methods are also much clearer. I have a couple of final, very minor comments/suggestions.

- “comorbid attentional atypicalities”: remove ‘comorbid’ since this does not relate to a clinical condition

- Table 2/Fig 3: typo in “Stimulus”

- “challenging the notion that a primitive sensitivity to protoface stimuli is essential for typical social development”: I think the authors meant ‘sufficient’ rather than ‘essential’ (the terms have different meanings). It also seems that the following sentence aims to express the same idea.

Response: We thanks Reviewer 1 for appreciating the current version of the ms. We fixed the typos accordingly. Changes are marked with the Track changes function of MS word in the ms.

- “whether such differences only emerge when attention is focused on the face”: it may be useful to clarify ‘attention is focused on the face’ since this could be interpreted in different ways depending on a reader’s academic background and their interpretation of 'attentional focus'

Response: We clarified the sentence as follow: “[…] whether such differences only emerge when infants pay attention to face. […]”

 - Thank you for explaining the age-matching procedure. It sounds like individuals were matched on both age and sex (as also stated in the Discussion), in which case it should be added in the methods for clarification, i.e., in the sentence “Participants diagnosed with ASD and TD that meet the inclusion criteria were matched on chronological age.”

Response: We added this information in the methods as follow: “Participants diagnosed with ASD and TD that meet the inclusion criteria were matched on chronological age and sex.”

- Thank you for clarifying the quality criteria. It seems the authors adopted my phrasing (“time taken until entry the AOI that contained the target”), but in the context of the ms I would suggest something along the lines of: ‘d) participants’ gaze entered the AOI that contained the target’ - I assume this is what was meant.

Response: Thank you, we revised the sentence accordingly.

- The updated figures are much easier to interpret and highlight the findings, thank you. Are Fig 3 and 4 displaying the same dataset but as individual points (Fig3) vs summary stats + error bars (Fig4)? If yes, I wonder if the graphs can be combined, although not a necessity. Individual points and summary stats are both useful, but if figures show the same data then it could be confusing as to why separate figures were generated.

Response: We appreciated the suggestion of the Reviewer 1. We combined Fig3 and Fig4 in a unique figure (i.e. Fig.3). We combined the figure’ captions accordingly by indicating two separate sections i.e., a) and b).

- “reduce confusion in the estimation…”: not clear what is meant by this part

Response: We clarify the meaning of the sentence as follow: “This evidence stresses the importance of carefully choosing non-social stimuli to clearly disambiguate between the impact of social and non-social stimuli on attention, in individuals with ASD.”

- “collide with”: contrast?

- “we found a quite remarkable heterogeneity”: for the ms, it might be better written as “we observed greater variability in the data from the ASD group compared to the TD group”, or similar.

- “huge variability”: replace huge with ‘high’

- “Continuos”: typo, ‘continuous’

Response: We thank the Reviewer 1 to carefully going through stylistic issues in the previous version of the ms. We corrected the submitted ms revision accordingly.

Reviewer #2: The manuscript is greatly improved. The introduction now reads clearly, and better sets up the premise of the experiment; particularly, the discussion on space-based vs object-based attention and why Egly’s paradigm is a suitable choice for this study. The methods and accompanying figures are more intelligible. The discussion better highlights the implications of the present set of results.

My comments are as follows:

1. Page 2, ref [20] and [21]: “On the one hand, to explain why people with ASD spend less time with attention on faces, some authors have proposed that this might derive from an innate atypicality of the face detection mechanism, that is the subcortical mechanism that tunes infant attention to face-like stimuli from birth [20]. However, evidence indicates that individuals with ASD exhibit entirely typical orienting responses to face-like stimuli, challenging the notion that a primitive sensitivity to protoface stimuli is essential for typical social development [21].”

Please note that reference [20] (Shah, Gaule, Bird, & Cook 2013) proposes that the robust orienting effect towards proto-face stimuli in the ASD population which they find speaks against developmental accounts which suggest that reduced looking behaviour in ASD is a consequence of atypical face detection mechanisms. Accordingly, please cite appropriate references to support the first sentence and please include reference [20] along with reference [21].

Response: We thank the Reviewer 2 to pointing out this reference typo. We modified the reference list by substituting the reference [20] with the study by Di Giorgio, et al.. (2016). Difference in visual social predispositions between newborns at low-and high-risk for autism. Scientific reports, 6(1), 1-9.

We update the references list accordingly, hence in the revised ms the study by Shah, Gaule, Bird, & Cook 2013 refers to [21] as in the sentence: “However, evidence indicates that individuals with ASD exhibit entirely typical orienting responses to face-like stimuli, challenging the notion that a primitive sensitivity to protoface stimuli is essential for typical social development [21]”.

2. Page 2: “Accordingly, several studies reported that individuals with a diagnosis of ASD show comorbid attentional atypicalities involving the alerting network”.

Attentional difficulties are not necessarily “comorbid” with ASD in the way that another disorder would be. Please remove “comorbid”, the sentence will then read: … “individuals with a diagnosis of ASD show attentional atypicalities involving the alerting network”.

Response: We removed the term “comorbid” accordingly, thank you. Changes are marked in the text with the Track changes function of MS word.

3. Page 3: “To answer these questions, we employed a paradigm suitable for evaluating costs and benefits in dislocating attention toward different attentional focuses.”

The authors make the argument for space-based and object-based attention lying on a continuum. In which case, might it be more suitable to replace “dislocating” with “deploying” or another similar word.

Response: We corrected the revised ms as suggested, thank you.

4. Page 3: “That is, space-based and object-based attention lie on a continuum rather than be different discrete components of the orienting of attention.”

Instead of “be”, “being” might read better.

Response: We corrected the revised ms as suggested, thank you.

5. Page 3, last para: “However, unlike space based attention, only object-based attention is a gateway to investigating the complex interplay [of which processes?] underling the impact of the selected information (object representation) on the deployment of attentional strategies”.

“Complex interplay” implies that two or more processes are involved, which is not made clear currently. Is “underling” perhaps “underlying”?

Response: We thank the Reviewer 2 for the careful reading of the manuscript, we modified the sentence as follow: “However, unlike space-based attention, only object-based attention is a gateway to investigating the impact of the selected information (object representation) on the deployment of attentional strategies [41].”

6. Page 4, present study: “In the current study we compared [which groups?]”.

It would be good to state which sample groups were included at the beginning.

Response: We clarify the sentence as follow: “In the current study, we compared the ability to shift attention both toward spatial locations (space-based attention) and objects (object-based attention) in children with, and without, a diagnosis of ASD.”

7. Page 9, data analysis: “time taken until entry [to] the AOI that contained the target”.

Please include the missing word “to”.

Response: We corrected the revised ms as suggested by the Reviewer 1, as follow: “(d) participants’ gaze entered the AOI that contained the target.”

8. Page 11: “However, in contrast to the TD group, in the ASD group no cost emerged in attention shift between- vs within-objects” [for both face and phase-scrambled stimuli].

Perhaps this could this be added to the sentence to summarise results better?

Response: We corrected the revised ms as suggested, thank you.

9. Page 13, discussion: “Crucially, [in which trials?], stimulus processing is not influenced by the type of the stimulus, but only by the correspondence between the position of the cue and those of the target.

Please could the authors specify in which trials this correspondence is expected to influence performance, i.e., is this the case for only valid trials or all trials facilitated by space-based attention? Presumably the latter, but this would need to be made clear please.

Response: We thank the Reviewer 2 for the suggestion. We tried to better explain this sentence as follow: “Crucially in all conditions stimulus processing is not influenced by the type of the stimulus, but only by the position of the cue and that of the target”.

---

## [Decision Letter · Decision Letter 2]

8 Apr 2021

PONE-D-20-35397R2

Attentional shift within and between faces: evidence from children with and without a diagnosis of autism spectrum disorder

PLOS ONE

Dear Dr. Calignano,

Thank you for submitting your manuscript to PLOS ONE. After careful consideration, we feel that it has merit but does not fully meet PLOS ONE’s publication criteria as it currently stands. Therefore, we invite you to submit a revised version of the manuscript that addresses the points raised during the review process.

As you can see, both reviewers are happy with your revisions, but one reviewer raised a handful of minor comments. I believe it would be straightforward to implement these suggestions, after which I'm hopeful to accept the paper.

We look forward to receiving your revised manuscript.

Kind regards,

Atsushi Senju

Academic Editor

PLOS ONE

Journal Requirements:

Reviewers' comments:

Reviewer's Responses to Questions

**Comments to the Author**

1. If the authors have adequately addressed your comments raised in a previous round of review and you feel that this manuscript is now acceptable for publication, you may indicate that here to bypass the “Comments to the Author” section, enter your conflict of interest statement in the “Confidential to Editor” section, and submit your "Accept" recommendation.

Reviewer #1: (No Response)

Reviewer #2: All comments have been addressed

2. Is the manuscript technically sound, and do the data support the conclusions?

Reviewer #1: Yes

Reviewer #2: Yes

3. Has the statistical analysis been performed appropriately and rigorously? 

Reviewer #1: I Don't Know

Reviewer #2: Yes

4. Have the authors made all data underlying the findings in their manuscript fully available?

Reviewer #1: Yes

Reviewer #2: Yes

5. Is the manuscript presented in an intelligible fashion and written in standard English?

Reviewer #1: Yes

Reviewer #2: Yes

6. Review Comments to the Author

Reviewer #1: The authors have responded to my comments in a satisfactory manner. I have one last, very minor comment and otherwise only noted down some typos (it may be useful to do a final proof-read).

Comment: p.3, ‘emerge when infants pay attention to face’ - should be corrected to 'to the face'. Secondly, I am not sure why the authors revised this to refer to infants specifically. I would replace with ‘individuals’ or 'children' if this is what the authors meant.

Typos:

- p.2: “confirm these pattern of data” change to “confirm these patterns of data”

- p.4: “rest of the object benefit” change to “rest of the object benefits”

- p.6: “accordance with the principles laid down in the Declaration of Helsinki” change to “accordance with the Declaration of Helsinki”

- p.9: “detection time for each trials” change to “detection time for each trial”

- p.13: “but only by the the position”; delete 1x ‘the’.

Reviewer #2: (No Response)

7. PLOS authors have the option to publish the peer review history of their article (what does this mean?). If published, this will include your full peer review and any attached files.

Reviewer #1: No

Reviewer #2: No

---

## [Author Response · Author response to Decision Letter 2]

13 Apr 2021

Reviewer #1: The authors have responded to my comments in a satisfactory manner. I have one last, very minor comment and otherwise only noted down some typos (it may be useful to do a final proof-read).

Comment: p.3, ‘emerge when infants pay attention to face’ - should be corrected to 'to the face'. Secondly, I am not sure why the authors revised this to refer to infants specifically. I would replace with ‘individuals’ or 'children' if this is what the authors meant.

Typos:

- p.2: “confirm these pattern of data” change to “confirm these patterns of data”

- p.4: “rest of the object benefit” change to “rest of the object benefits”

- p.6: “accordance with the principles laid down in the Declaration of Helsinki” change to “accordance with the Declaration of Helsinki”

- p.9: “detection time for each trials” change to “detection time for each trial”

- p.13: “but only by the the position”; delete 1x ‘the’.

Authors: We thank Reviewer 1, we carefully edited the ms as suggested. Please find all changes marked by the track-changes function in the revised ms.

---

## [Editor Report · Decision Letter 3]

28 Apr 2021

Attentional shift within and between faces: evidence from children with and without a diagnosis of autism spectrum disorder

PONE-D-20-35397R3

Dear Dr. Calignano,

We’re pleased to inform you that your manuscript has been judged scientifically suitable for publication and will be formally accepted for publication once it meets all outstanding technical requirements.

Kind regards,

Atsushi Senju

Academic Editor

PLOS ONE
---

## [Editor Report · Acceptance letter]

4 May 2021

PONE-D-20-35397R3 

Attentional shift within and between faces: evidence from children with and without a diagnosis of autism spectrum disorder 

Dear Dr. Calignano:

I'm pleased to inform you that your manuscript has been deemed suitable for publication in PLOS ONE. Congratulations! Your manuscript is now with our production department. 

Kind regards, 

on behalf of

Dr. Atsushi Senju 

Academic Editor

PLOS ONE